# DECOUPLED KULLBACK-LEIBLER DIVERGENCE LOSS

## ABSTRACT

In this paper, we delve deeper into the Kullback–Leibler (KL) Divergence loss and mathematically prove that it is equivalent to the Doupled Kullback-Leibler (DKL) Divergence loss that consists of 1) a weighted Mean Square Error ($w$MSE) loss and 2) a Cross-Entropy loss incorporating soft labels. From our analysis of the DKL loss, we have identified two areas for improvement. Firstly, we address the limitation of DKL in scenarios like knowledge distillation by breaking its asymmetry property in training optimization. This modification ensures that the $w$MSE component is always effective during training, providing extra constructive cues. Secondly, we introduce global information into DKL to mitigate bias from individual samples. With these two enhancements, we derive the Improved Kullback–Leibler (IKL) Divergence loss and evaluate its effectiveness by conducting experiments on CIFAR-10/100 and ImageNet datasets, focusing on adversarial training and knowledge distillation tasks. The proposed approach achieves new state-of-the-art adversarial robustness on the public leaderboard — *R*obustBench and competitive performance on knowledge distillation, demonstrating the substantial practical merits.

## 1 INTRODUCTION

Loss functions are a critical component of training deep models. Cross-Entropy loss is particularly important in image classification tasks He et al. (2016); Simonyan & Zisserman (2015); Tan et al. (2019); Dosovitskiy et al. (2020); Liu et al. (2021), while Mean Square Error (MSE) loss is commonly used in regression tasks Ren et al. (2015); He et al. (2017; 2022). Contrastive loss Chen et al. (2020a); He et al. (2020); Chen et al. (2020b); Grill et al. (2020); Caron et al. (2020); Cui et al. (2021b; 2022) has emerged as a popular objective for representation learning. The selection of an appropriate loss function can exert a substantial influence on a model's performance. Therefore, the development of effective loss functions Cao et al. (2019); Lin et al. (2017); Zhao et al. (2022); Wang et al. (2020); Johnson et al. (2016); Berman et al. (2018); Wen et al. (2016); Tan et al. (2020) remains a critical research topic in the fields of computer vision and machine learning.

Kullback-Leibler (KL) Divergence quantifies the degree of dissimilarity between a probability distribution and a reference distribution. As one of the most frequently used loss functions, it finds application in various scenarios, such as adversarial training Zhang et al. (2019); Wu et al. (2020); Cui et al. (2021a); Jia et al. (2022), knowledge distillation Hinton et al. (2015); Chen et al. (2021); Zhao et al. (2022), incremental learning Chaudhry et al. (2018); Lee et al. (2017), and robustness on out-of-distribution data Hendrycks et al. (2019). Although many of these studies incorporate KL Divergence loss as part of their algorithms, they may not thoroughly investigate the underlying mechanisms of the loss function. To address this issue, our paper aims to elucidate the working mechanism of KL Divergence with respect to gradient optimization.

Our study focuses on the Kullback–Leibler (KL) Divergence loss from the perspective of gradient optimization. We provide theoretical proof that it is equivalent to the Decoupled Kullback–Leibler (DKL) Divergence loss, which comprises a weighted Mean Square Error ($w$MSE) loss and a Cross-Entropy loss with soft labels. We have identified potential issues with the DKL loss. Specifically, its gradient optimization is asymmetric with respect to inputs, which can lead to the weighted MSE ($w$MSE) component being ignored in certain scenarios, such as knowledge distillation. Fortunately, it is convenient to address this issue with the new formulation of DKL by breaking the asymmetry property. Moreover, global information is used to regularize the training process as a holistic categorical distribution prior. Combining DKL with these two points, we derive the Improved Kull-

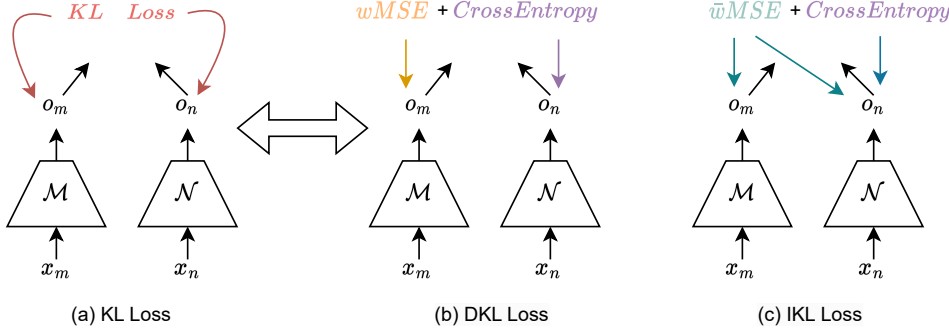

Figure 1: **Comparison of gradient backpropagation among KL, DKL, and IKL losses.** $\mathcal{M}$ and $\mathcal{N}$ can be the same one or two separate models determined by application scenarios. Similarly, $x_m$, $x_n \in \mathcal{X}$ can also be the same one or two different images. $o_m$, $o_n$ are logits output with which the probability vector can be obtained when applying the *Softmax* activation. Black arrows represent the forward process while colored arrows indicate the gradient backpropagation flows in the backward process driven by the corresponding loss functions in the same color. "$wMSE$" is a weighted MSE loss. "$\bar{w}MSE$" is incorporated with global information.

back–Leibler (IKL) Divergence loss. Figure 1 presents a clear visual comparison of the KL, DKL, and IKL loss functions in terms of gradient optimization during backpropagation.

To demonstrate the effectiveness of our proposed IKL loss, we evaluate it in adversarial training and knowledge distillation tasks. Our experimental results on CIFAR-10/100 and ImageNet show that the IKL loss achieves new state-of-the-art robustness on the public leaderboard of *R*obustBench [1]. In summary, the main contributions of our work are:

- Our study provides insights into the Kullback–Leibler (KL) Divergence loss by analyzing its gradient optimization properties. In doing so, we reveal that it is mathematically equivalent to a combination of a weighted Mean Square Error ($w$MSE) loss and a Cross-Entropy loss with soft labels.

- After analyzing the Decoupled Kullback-Leibler (DKL) Divergence loss, we propose two modifications for enhancement: addressing its asymmetry property and incorporating global information. The derived Improved Kullback-Leibler (IKL) Divergence loss demonstrates improved performance.

- By utilizing the IKL loss for adversarial training and knowledge distillation, we obtain the state-of-the-art adversarial robustness on *R*obustBench and competitive knowledge distillation performance on CIFAR-10/100 and ImageNet.

## 2 RELATED WORK

**Adversarial Robustness.** Since the identification of adversarial examples by Szegedy et al. Szegedy et al. (2013), the security of deep neural networks (DNNs) has gained significant attention, and ensuring the reliability of DNNs has become a prominent topic in the deep learning community. Numerous algorithms have been developed to defend against adversarial attacks. However, as highlighted by Athalye et al. Athalye et al. (2018), methods relying on obfuscated gradients can create a deceptive sense of security and remain vulnerable to strong attacks such as auto-attack Croce & Hein (2020). Adversarial training Madry et al. (2017), being the most effective method, stands out due to its consistently high performance.

Adversarial training incorporates adversarial examples into the training process. Madary et al. Madry et al. (2017) propose the adoption of the universal first-order adversary, specifically the PGD attack, in adversarial training. Zhang et al. Zhang et al. (2019) enhance model robustness by utilizing the Kullback-Leibler (KL) Divergence loss based on their theoretical analysis. Wu et al. Wu et al. (2020) introduce adversarial weight perturbation to explicitly regulate the flatness of the weight loss landscape. Cui et al. Cui et al. (2021a) leverage guidance from naturally-trained models

---

[1]https://robustbench.github.io/

to regularize the decision boundary in adversarial training. Additionally, various other techniques Jia et al. (2022) focusing on optimization or training aspects have also been developed.

In recent years, several works have explored the use of data augmentation techniques to improve adversarial training. Gowal et al. Gowal et al. (2021) have shown that synthesized images using generative models can enhance adversarial training and improve robustness against adversarial attacks. Wang et al. Wang et al. (2023) have demonstrated that stronger robustness can be achieved by utilizing better generative models such as the popular diffusion model Karras et al. (2022), resulting in new state-of-the-art adversarial robustness. Additionally, Addepalli et al. Addepalli et al. (2022) have made it feasible to incorporate general augmentation techniques for image classification, such as Autoaugment Cubuk et al. (2019) and CutMix Yun et al. (2019a), into adversarial training.

We have explored the mechanism of KL loss for adversarial robustness in this paper. The effectiveness of the proposed IKL loss is tested in both settings with and without synthesized data.

**Knowledge Distillation.** The concept of Knowledge Distillation (KD) was first introduced by Hinton et al. Hinton et al. (2015). It involves extracting "dark knowledge" from accurate teacher models to guide the learning process of student models, which often have lower capacity than their teachers. This is achieved by utilizing the Kullback-Leibler Divergence (KL) loss to regularize the output probabilities of student models, aligning them with those of their teacher models when given the same inputs. This simple yet effective technique significantly improves the generalization ability of smaller models and finds extensive applications in various domains. Since the initial success of KD Hinton et al. (2015), several advanced methods, including logits-based Cho & Hariharan (2019); Furlanello et al. (2018); Mirzadeh et al. (2020); Yang et al. (2019); Zhang et al. (2018); Zhao et al. (2022); Huang et al. (2022) and features-based approaches Romero et al. (2015); Tian et al. (2020); Heo et al. (2019b); Zagoruyko & Komodakis (2017); Chen et al. (2021); Heo et al. (2019a;c); Kim et al. (2018); Park et al. (2019); Peng et al. (2019); Yim et al. (2017), have been introduced.

Logits-based methods extract only the logits output from teacher models. These methods are more general than features-based methods as there is no requirement to know the teacher model architecture, and only the logits output is needed for inputs. Features-based methods explore to take advantage of intermediate layer features compared with logits-based methods. This kind of method usually requires knowing the architecture of teacher models. With such extra priors and features information, features-based methods are expected to achieve higher performance, which can be along with more computation or storage costs.

This paper decouples the KL loss into a new formulation, *i.e.*, DKL, and addresses the limitation of KL loss for application scenarios like knowledge distillation. With the improved version of DKL, *i.e.*, IKL loss, our models even surpass all previous features-based methods.

## 3 METHOD

In this section, we detail the preliminary and our motivation in Section 3.1, and then discuss our Improved Kullback-Leibler (IKL) Divergence loss in Section 3.2.

### 3.1 PRELIMINARY AND MOTIVATION

**Revisiting Kullback-Leibler (KL) Divergence Loss.** Kullback-Leibler (KL) Divergence measures the differences between two probability distributions. For distributions $P$ and $Q$ of a continuous random variable, It is defined to be the integral:

$$D_{KL}(P||Q) = \int_{-\infty}^{+\infty} p(x) * \log \frac{p(x)}{q(x)} dx, \tag{1}$$

where $p$ and $q$ denote the probability densities of $P$ and $Q$.

KL loss is one of the most commonly used objectives in deep learning. In this paper, we study the mechanism of KL loss and test our Improved Kullback-Leibler (IKL) Divergence loss with adversarial training and knowledge distillation tasks. For adversarial training, to enhance model robustness, KL loss regularizes the output probability vector of adversarial examples to be the same as that of their corresponding clean images. Knowledge distillation algorithms adopt KL loss to let a student model mimic behaviors of one teacher model. With the transferred knowledge from the teacher, the student is expected to improve performance.

**Preliminaries.** We consider image classification models that predict probability vectors with the *Softmax* activation. Assume $o_i \in \mathcal{R}^C$ is the logits output of one deep model with an image $x_i \in \mathcal{X}$ as input, where $C$ is the number of classes in the task. $s_i \in \mathcal{R}^C$ is the predicted probability vector and $s_i = \textit{Softmax}(o_i)$. $o_i^j$ and $s_i^j$ are values for the $j$-th class in $o_i$ and $s_i$ respectively.

KL loss is applied to make $s_m$ and $s_n$ similar in many scenarios, leading to the following objective,

$$\mathcal{L}_{KL}(x_m, x_n) = \sum_{j=1}^{C} s_m^j * \log \frac{s_m^j}{s_n^j}. \tag{2}$$

For instance, in adversarial training, $x_m$ is a natural image, and $x_n$ is the corresponding adversarial example of $x_m$. $x_m$ and $x_n$ indicate the same image and are fed into the teacher and student models separately in knowledge distillation. It is worth noting that $s_m$ is detached from the gradient backpropagation because the teacher model is well-trained and fixed in the distillation process.

**Motivation.** Previous works Hinton et al. (2015); Zhao et al. (2022); Zhang et al. (2019); Cui et al. (2021a) incorporate the KL loss into their algorithms without exploring its inherent working mechanism. The objective of this paper is to uncover the driving force behind training optimization through an examination of the KL loss function. With the back-propagation rule, the derivative gradients are as follows,

$$\frac{\partial \mathcal{L}_{KL}}{\partial o_m^j} = \sum_{k=1}^{C}((\Delta m_{j,k} - \Delta n_{j,k}) * (s_m^k * s_m^j)), \tag{3}$$

$$\frac{\partial \mathcal{L}_{KL}}{\partial o_n^j} = s_m^j * (s_n^j - 1) + s_n^j * (1 - s_m^j), \tag{4}$$

where $\Delta m_{j,k} = o_m^j - o_m^k$, and $\Delta n_{j,k} = o_n^j - o_n^k$.

Taking advantage of the antiderivative technique with such gradient information, we introduce a novel formulation - the Decoupled Kullback-Leibler (DKL) Divergence loss - which is presented in Lemma 1. The DKL loss is expected to be equivalent to the KL loss and prove to be a more analytically tractable alternative for further exploration and study.

**Lemma 1** From the perspective of gradient optimization, the Kullback-Leibler (KL) Divergence loss is equivalent to the following Decoupled Kullback-Leibler (DKL) Divergence loss when $\alpha = 1$ and $\beta = 1$.

$$\mathcal{L}_{DKL}(x_m, x_n) = \underbrace{\frac{\alpha}{4} \sum_{j=1}^{C} \sum_{k=1}^{C}((\Delta m_{j,k} - \mathcal{S}(\Delta n_{j,k}))^2 * \mathcal{S}(w_m^{j,k}))}_{\textbf{weighted MSE } (w\textbf{MSE})} - \underbrace{\beta \sum_{j=1}^{C} \mathcal{S}(s_m^j) * \log s_n^j}_{\textbf{Cross-Entropy}}, \tag{5}$$

where $\mathcal{S}(\cdot)$ means *stop gradients* operation. $w_m^{j,k} = s_m^j * s_m^k$.

As demonstrated by Lemma 1 and Section 3.1, we can conclude the following key properties of KL and DKL.

- DKL loss is equivalent to KL loss in terms of gradient optimization. Thus, KL loss can be decoupled into a weighted Mean Square Error ($w$MSE) loss and a Cross-Entropy loss incorporating soft labels.

- Optimization is asymmetric for $o_m$ and $o_n$. The $w$MSE and Cross-Entropy losses in Equation (5) are complementary and collaboratively work together. The asymmetry property can cause the $w$MSE to be neglected or overlooked when $o_m$ is detached from gradient backpropagation, which is the case for knowledge distillation.

- "$w_m^{j,k}$" in Equation (5) is conditioned on the prediction of $x_m$. Nevertheless, sample-wise predictions may be subject to significant variance. Incorrect predictions for hard examples can result in unstable training and challenging optimization problems.

### 3.2 IMPROVED KULLBACK-LEIBLER (IKL) DIVERGENCE LOSS

Based on the analysis in Section 3.1, we propose an Improved Kullback-Leibler (IKL) Divergence loss. Compared with DKL in Equation (5), we make the following improvements: 1) **breaking the asymmetry property**; 2) **introducing global information**. The details are presented as follows.

**Breaking the Asymmetry Property.** As shown in Equation (5), the weighted MSE ($w$MSE) encourages $o_n$ to be similar to $o_m$ with the second-order information, *i.e.*, logit differences between any two classes. The cross-entropy loss guarantees that $s_n$ can have the same predicted scores with $s_m$. Two loss terms collaboratively work together to make $o_n$ and $o_m$ similar absolutely and relatively. Discarding any one of them can lead to performance degradation.

However, because of the asymmetry property of KL/DKL, the unexpected case may occur when $s_m$ is detached from the gradient back-propagation (scenarios like knowledge distillation), which is formulated as:

$$\mathcal{L}_{DKL-KD}(x_m, x_n) = \underbrace{\frac{\alpha}{4} \sum_{j=1}^{C} \sum_{k=1}^{C} ((\mathcal{S}(\Delta m_{j,k}) - \mathcal{S}(\Delta n_{j,k}))^2 * \mathcal{S}(w_m^{j,k}))}_{\text{weighted MSE } (w\text{MSE})} - \underbrace{\beta \sum_{j=1}^{C} \mathcal{S}(s_m^j) * \log s_n^j}_{\text{Cross-Entropy}},$$

(6)

where $\mathcal{S}(\cdot)$ means *stop gradients* operation. $w_m^{j,k} = s_m^j * s_m^k$.

As indicated by Equation (6), the weighted MSE ($w$MSE) loss will take no effect on training optimization since all components of $w$MSE are detached from gradient propagation, which can potentially hurt the model performance. Knowledge distillation matches this case because the teacher model is fixed during distillation training.

We address this issue by breaking the asymmetry property of KL/DKL, *i.e.*, enabling the gradients of $\mathcal{S}(\Delta n_{j,k})$. The updated formulation becomes,

$$\widehat{\mathcal{L}}_{DKL-KD}(x_m, x_n) = \underbrace{\frac{\alpha}{4} \sum_{j=1}^{C} \sum_{k=1}^{C} ((\mathcal{S}(\Delta m_{j,k}) - \Delta n_{j,k})^2 * \mathcal{S}(w_m^{j,k}))}_{\text{weighted MSE } (w\text{MSE})} - \underbrace{\beta \sum_{j=1}^{C} \mathcal{S}(s_m^j) * \log s_n^j}_{\text{Cross-Entropy}}, \quad (7)$$

where $\mathcal{S}(\cdot)$ means *stop gradients* operation. $w_m^{j,k} = s_m^j * s_m^k$.

**Introducing Global Information.** The *weights* for the weighted MSE ($w$MSE) of DKL in Equation (5) is sample-wise and depends on the prediction $s_m$,

$$w_m^{j,k} = s_m^j * s_m^k. \tag{8}$$

However, sample-wise *weights* can be biased due to the individual prediction variance. Specifically, models can give wrong predictions for hard examples, leading to challenging optimization. We thus adopt class-wise *weights* for IKL loss,

$$\bar{w}_y^{j,k} = \bar{s}_y^j * \bar{s}_y^k, \tag{9}$$

where $y$ is ground-truth label of $x_m$, $\bar{s}_y = \frac{1}{|\mathcal{X}_y|} \sum_{x_i \in \mathcal{X}_y} s_i$.

The global information injected by $\bar{w}_y^{j,k}$ can act as a regularization to enhance intra-class consistency and mitigate biases that may arise from sample noise.

To this end, we derive the IKL loss in Equation (10) by incorporating these two designs,

$$\mathcal{L}_{IKL}(x_m, x_n) = \underbrace{\frac{\alpha}{4} \sum_{j=1}^{C} \sum_{k=1}^{C} ((\Delta m_{j,k} - \Delta n_{j,k})^2 * \mathcal{S}(\bar{w}_y^{j,k}))}_{\text{global weighted MSE } (\bar{w}\text{MSE})} - \underbrace{\beta \sum_{j=1}^{C} \mathcal{S}(s_m^j) * \log s_n^j}_{\text{Cross-Entropy}}, \tag{10}$$

where $y$ is the ground-truth label for $x_m$. $\bar{w}_y \in \mathcal{R}^{C \times C}$ is the weights for class $y$.

Table 1: **Ablation study on "GI" and "BA" with DKL loss.** "GI" is "Global Information", and "BA" indicates "Breaking Asymmetry". "Clean" is the test accuracy of clean images and "AA" is the Robustness under AutoAttack. CIFAR-100 is used for adversarial training and ImageNet is adopted for knowledge distillation.

| Index | GI | BA | Adversarial Training Clean (%) | AA (%) | Knowledge Distillation Top-1 (%) | Descriptions |
|-------|----|----|------------------|--------|---------------------------|--------------|
| (a) | Na | Na | 62.87 | 30.29 | 71.03 | baseline with KL loss. |
| (b) | ✗ | ✗ | 62.54 | 30.20 | 71.03 | DKL, equivalent to KL loss. |
| (c) | ✗ | ✔ | 62.69 | 30.42 | 71.80 | (b) with BA. |
| (d) | ✔ | ✔ | 66.51 | **31.45** | **71.91** | (c) with GI, *i.e.*, IKL. |

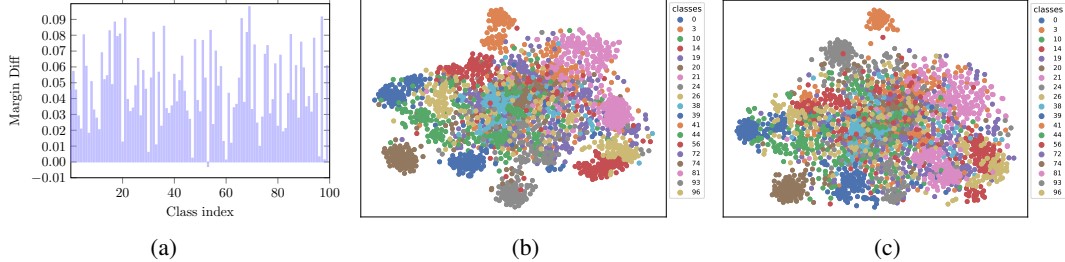

| (a) | (b) | (c) |

Figure 2: **Visualization comparisons.** (a) Class margin differences between models trained by IKL-AT and TRADES; (b) t-SNE visualization of the model trained by IKL-AT on CIFAR-100; (c) t-SNE visualization of the model trained by TRADES on CIFAR-100.

### 3.3 A CASE STUDY AND ANALYSIS

**A Case Study.** We empirically examine each component of IKL on CIFAR-100 with the adversarial training task and on ImageNet with the knowledge distillation task. Ablation experimental results and their setting descriptions are listed in Table 1. In the implementation, for adversarial training, we use improved TRADES Zhang et al. (2019) as our baseline that combines with AWP Wu et al. (2020) and uses an increasing epsilon schedule Addepalli et al. (2022). For knowledge distillation, we use the official code from DKD. The comparison between (a) and (b) shows that DKL can achieve comparable performance, confirming the equivalence to KL. The comparisons among (b), (c), and (d) validate the effectiveness of the "GI" and "BA" mechanisms.

**Analysis on Introducing Global Information.** As demonstrated by Table 1, global information plays an important role in adversarial robustness. The mean probability vector $\bar{p}_y$ of all samples in the class $y$ is more robust than the sample-wise probability vector. During training, once the model gives incorrect predictions for hard samples, $w_m^{j,k}$ in Equation (5) will wrongly guide the optimization and make the training unstable and challenging.

Adoption of $\bar{w}_y^{j,k}$ in Equation (6) can mitigate the issue and meanwhile enhance intra-class consistency. Especially, in the late training stage, most of the training samples are fitted well and the class-wise probability vector always gives us correct guidance in optimization. To visualize the effectiveness of introducing global information, we define the boundary margin for class $y$ as:

$$\text{Margin}_y = \bar{p}_y[y] - \max_{k!=y} \bar{p}_y[k]. \tag{11}$$

We plot the margin differences between models trained by IKL-AT and TRADES on CIFAR-100. As shown in Figure 2(a), almost all class margin differences are positive, demonstrating that there are larger decision boundary margins for the IKL-AT model. Such larger margins lead to stronger robustness. This phenomenon is coherent with our experimental results in Section 4.1.

We also randomly sample 20 classes in CIFAR-100 for t-SNE visualization. The numbers in the pictures are class indexes. For each sampled class, we collect the feature representation of natural images and adversarial examples with the validation set. The visualization by t-SNE is shown in Figure 2(c) and Figure 2(b). Compared with TRADES that trained with KL loss, Features by IKL-AT models are more compact and separable.

Table 2: **Test accuracy (%) of clean images and Robustness (%) under AutoAttack on CIFAR-100.** All results are the average over three trials.

| Dataset | Method | Architecture | Augmentation Type | Clean | AA |
|---|---|---|---|---|---|
| | AWP | WRN-34-10 | Basic | 60.38 | 28.86 |
| | LBGAT | WRN-34-10 | Basic | 60.64 | 29.33 |
| | LAS-AT | WRN-34-10 | Basic | 64.89 | 30.77 |
| | ACAT | WRN-34-10 | Basic | 65.75 | 30.23 |
| **CIFAR-100** | **IKL-AT** | WRN-34-10 | Basic | **66.51** | **31.45** |
| $(\ell_\infty, \epsilon = 8/255)$ | | WRN-34-10 | Basic | 63.40 | **31.92** |
| | ACAT | WRN-34-10 | AutoAug | **68.74** | 31.30 |
| | **IKL-AT** | WRN-34-10 | AutoAug | 65.93 | **32.52** |
| | Wang et al. (2023) | WRN-28-10 | 50M Generated Data | 72.58 | 38.83 |
| | **IKL-AT** | WRN-28-10 | 50M Generated Data | **73.85** | **39.18** |

Table 3: **Top-1 accuracy (%) on the ImageNet validation and training speed (sec/iteration) comparisons.** Training speed is calculated on 4 Nvidia GeForce 3090 GPUs with a batch of 512 224x224 images. All results are the average over three trials.

| Distillation Manner | Teacher / Student | Extra Parameters | ResNet34 73.31 / ResNet18 69.75 | | ResNet50 76.16 / MobileNet 68.87 | |
|---|---|---|---|---|---|---|
| | AT | ✗ | 70.69 | | 69.56 | |
| Features | OFD | ✔ | 70.81 | | 71.25 | |
| | CRD | ✔ | 71.17 | | 71.37 | |
| | ReviewKD | ✔ | 71.61 | 0.319 s/iter | 72.56 | 0.526 s/iter |
| | DKD | ✗ | 71.70 | | 72.05 | |
| Logits | KD | ✗ | 71.03 | | 70.50 | |
| | **IKL-KD** | ✗ | **71.91** | **0.197 s/iter** | **72.84** | **0.252 s/iter** |

# 4 EXPERIMENTS

To verify the effectiveness of the proposed IKL loss, we conduct experiments on CIFAR-10, CI-FAR100, and ImageNet for adversarial training (Section 4.1) and knowledge distillation (Section 4.2).

## 4.1 ADVERSARIAL ROBUSTNESS

**Experimental Settings.** We use an improved version of TRADES Zhang et al. (2019) as our baseline, which incorporates AWP Wu et al. (2020) and adopts an increasing epsilon schedule Addepalli et al. (2022). SGD optimizer with a momentum of 0.9 is used. We use the cosine learning rate strategy with an initial learning rate of 0.2 and train models 200 epochs. The batch size is 128, the weight decay is 5e-4 and the perturbation size $\epsilon$ is set to 8 / 255. Following previous work Zhang et al. (2019); Cui et al. (2021a), standard data augmentation including random crops with 4 pixels of padding and random horizontal flip is performed for data preprocessing. An interesting phenomenon is that IKL-AT is complementary to data augmentation strategies, like AutoAug, without any specific designs, which is different from the previous observation that the data augmentation strategy hardly benefits adversarial training Wu et al. (2020).

Under the setting of training with generated data, we strictly follow the training configurations in Wang et al. (2023) for fair comparisons. Our implementations are based on their open-sourced code. We only replace the KL loss with our IKL loss.

**Datasets and Evaluation.** CIFAR-10 and CIFAR-100 are the two most popular benchmarks in the adversarial community. The CIFAR-10 dataset consists of 60,000 $32 \times 32$ color images in 10 classes, with 6,000 images per class. There are 50,000 training images and 10,000 test images. The more challenging CIFAR-100 has 100 classes containing 600 images each. There are 500 training images and 100 testing images per class.

Following previous work Wu et al. (2020); Cui et al. (2021a), we report the clean accuracy on natural images and adversarial robustness under auto-attack Croce & Hein (2020) with epsilon 8/255.

Table 4: **Test accuracy (%) of clean images and robustness (%) under AutoAttack on CIFAR-10.** All results are the average over three trials.

| Dataset | Method | Architecture | Augmentation Type | Clean | AA |
|---|---|---|---|---|---|
| **CIFAR-10** ($\ell_\infty, \epsilon = 8/255$) | Rice et al. (2020) | WRN-34-20 | Basic | 85.34 | 53.42 |
| | LBGAT | WRN-34-20 | Basic | **88.70** | 53.57 |
| | AWP | WRN-34-10 | Basic | 85.36 | 56.17 |
| | LAS-AT | WRN-34-10 | Basic | 87.74 | 55.52 |
| | ACAT | WRN-34-10 | Basic | 82.41 | 55.36 |
| | **IKL-AT** | WRN-34-10 | Basic | 85.31 | **57.13** |
| | ACAT | WRN-34-10 | AutoAug | 88.64 | 57.05 |
| | **IKL-AT** | WRN-34-10 | AutoAug | 85.20 | **57.62** |
| | Wang et al. (2023) | WRN-28-10 | 20M Generated Data | 92.44 | 67.31 |
| | **IKL-AT** | WRN-28-10 | 20M Generated Data | 92.16 | **67.75** |

Table 5: **Top-1 accuracy (%) on the CIFAR-100 validation.** Teachers and students are in the **same** architectures. And $\Delta$ represents the performance improvement over the classical KD. All results are the average over 3 trials.

| Distillation Manner | | ResNet56 72.34 | ResNet110 74.31 | ResNet32×4 79.42 | WRN-40-2 75.61 | WRN-40-2 75.61 | VGG13 74.64 |
|---|---|---|---|---|---|---|---|
| | Teacher | | | | | | |
| | Student | ResNet20 69.06 | ResNet32 71.14 | ResNet8×4 72.50 | WRN-16-2 73.26 | WRN-40-1 71.98 | VGG8 70.36 |
| Features | FitNet | 69.21 | 71.06 | 73.50 | 73.58 | 72.24 | 71.02 |
| | RKD | 69.61 | 71.82 | 71.90 | 73.35 | 72.22 | 71.48 |
| | CRD | 71.16 | 73.48 | 75.51 | 75.48 | 74.14 | 73.94 |
| | OFD | 70.98 | 73.23 | 74.95 | 75.24 | 74.33 | 73.95 |
| | ReviewKD | 71.89 | 73.89 | 75.63 | 76.12 | **75.09** | 74.84 |
| Logits | DKD | **71.97** | 74.11 | 76.32 | 76.24 | 74.81 | 74.68 |
| | KD | 70.66 | 73.08 | 73.33 | 74.92 | 73.54 | 72.98 |
| | **IKL-KD** | 71.44 | **74.26** | **76.59** | **76.45** | 74.98 | **74.98** |

**Comparison Methods.** To compare with previous methods, We categorize them into two groups according to the different types of data preprocessing:

- Methods Wu et al. (2020); Cui et al. (2021a); Liu et al. (2019) with basic augmentation, *i.e.*, random crops and random horizontal flip.
- Methods Wang et al. (2023); Pang et al. (2022); Gowal et al. (2021) using augmentation with generative models or AutoAug Cubuk et al. (2019), CutMix Yun et al. (2019b).

**Comparisons with State-of-the-art on CIFAR-100.** On CIFAR-100, with the basic augmentations setting, we compare with AWP, LBGAT, LAS-AT, and ACAT. The experimental results are summarized in Table 2. Our WRN-34-10 models trained with IKL loss do a better trade-off between natural accuracy and adversarial robustness. With $\alpha = 20$ and $\beta = 3$, the model achieves 66.51% top-1 accuracy on natural images while 31.45% robustness under auto-attack. Combined with AutoAug, we obtain **32.52%** robustness, achieving new state-of-the-art under the setting without extra real or generated data.

We follow Wang et al. (2023) to take advantage of synthesized images generated by the popular diffusion models Karras et al. (2022). With 50M generated images, we create new state-of-the-art with WideResNet-28-10, achieving **73.85%** top-1 natural accuracy and **39.18%** adversarial robustness under auto-attack.

**Comparison with State-of-the-art on CIFAR-10.** Experimental results on CIFAR-10 are listed in Table 4, with the basic augmentation setting, our model achieves 85.31% top-1 accuracy on natural images and 57.13% robustness, outperforming AWP by 0.96% on robustness. With extra generated data, we improve the state-of-the-art by 0.44%, achieving **67.75%** robustness.

## 4.2 KNOWLEDGE DISTILLATION

**Datasets and Evaluation.** Following previous work Chen et al. (2021); Tian et al. (2020), we conduct experiments on CIFAR-100 Krizhevsky & Hinton (2009) and ImageNet Russakovsky et al.

(2015) to show the advantages of IKL on knowledge distillation. ImageNet Russakovsky et al. (2015) is the most challenging dataset for classification, which consists of 1.2 million images for training and 50K images for validation over 1,000 classes.

For evaluation, we report top-1 accuracy on CIFAR-100 and ImageNet validation. The training speed of different methods is also discussed.

**Experimental Settings.** We follow the experimental settings in Zhao et al. (2022) by Zhao et al. Our implementation for knowledge distillation is based on their open-sourced code.

Specifically, on CIFAR-100, we train all models for 240 epochs with a learning rate that decayed by 0.1 at the 150th, 180th, and 210th epoch. We initialize the learning rate to 0.01 for MobileNet and ShuffleNet, and 0.05 for other models. The batch size is 64 for all models. We train all models three times and report the mean accuracy.

On ImageNet, we use the standard training that trains the model for 100 epochs and decays the learning rate for every 30 epochs. We initialize the learning rate to 0.2 and set the batch size to 512.

For both CIFAR-100 and ImageNet, we consider the distillation among the architectures having the same unit structures, like ResNet56 and ResNet20, VGGNet13 and VGGNet8. On the other hand, we also explore the distillation among architectures made up of different unit structures, like WideResNet and ShuffleNet, VggNet and MobileNet-V2.

**Comparison Methods.** According to the information extracted from the teacher model in distillation training, knowledge distillation methods can be divided into two categories:

- Features-based methods Romero et al. (2015); Tian et al. (2020); Chen et al. (2021); Heo et al. (2019b). This kind of method makes use of features from different layers of the teacher model, which can need extra parameters and high training computational costs.

- Logits-based methods Hinton et al. (2015); Zhao et al. (2022). This kind of method only makes use of the logits output of the teacher model, which does not require knowing the architectures of the teacher model and thus is more general in practice.

**Comparison with State-of-the-art on CIFAR-100.** Experimental results on CIFAR-100 are summarized in Table 5 and Table 6 (in Appendix). Table 5 lists the comparisons with previous methods under the setting that the architectures of the teacher and student have the same unit structures. Models trained by IKL-KD can achieve comparable or better performance in all considered settings. Specifically, we achieve the best performance in 4 out of 6 training settings. Table 6 in Appendix shows the comparisons with previous methods under the setting that the architectures of the teacher and student have different unit structures.

**Comparison with State-of-the-art on ImageNet.** We empirically show the comparisons with other methods on ImageNet in Table 3. With a ResNet34 teacher, our ResNet18 achieves **71.91%** top-1 accuracy. With a ResNet50 teacher, our MobileNet achieves **72.84%** top-1 accuracy. Models trained by IKL-KD surpass all previous methods while saving **38%** and **52%** computation costs for ResNet34–ResNet18 and ResNet50–MobileNet distillation training respectively when compared with ReviewKD Chen et al. (2021).

## 5    CONCLUSION AND LIMITATION

In this paper, we have investigated the mechanism of Kullback-Leibler (KL) Divergence loss in terms of gradient optimization. Based on our analysis, we decouple the KL loss into a weighted Mean Square Error ($w$MSE) loss and a Cross-Entropy loss with soft labels. The new formulation is named Decoupled Kullback-Leibler (DKL) Divergence loss. To address the spotted issues of DKL, we make two improvements that break asymmetry property in optimization and incorporate global information, deriving the Improved Kullback-Leibler (IKL) Divergence loss. Experimental results on CIFAR-10/100 and ImageNet show that we create new state-of-the-art adversarial robustness and competitive performance on knowledge distillation, indicating the effectiveness of our IKL loss. KL loss has various applications. we consider it as future work to showcase the potential of IKL in other scenarios.

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
