# A APPENDIX

## A.1 PROOF TO LEMMA 1

To demonstrate that DKL in Equation (5) is equivalent to KL in Equation (1) for training optimization, we prove that DKL and KL have the same gradients given the same inputs.

For KL loss, we have the following derivatives according to the chain rule:

$$\frac{\partial s_m^i}{\partial o_m^i} = s_m^i * \sum_{j!=i}^{C} s_m^j,$$

$$\frac{\partial s_m^j}{\partial o_m^i} = -s_m^i * s_m^j,$$

$$\frac{\partial \mathcal{L}_{KL}}{\partial s_m^i} = \log s_m^i - \log s_n^i + 1,$$

$$\frac{\partial \mathcal{L}_{KL}}{\partial o_n^i} = s_m^i * (s_n^i - 1) + s_n^i * (1 - s_m^i)$$

$$\frac{\partial \mathcal{L}_{KL}}{\partial o_m^i} = \frac{\mathcal{L}_{KL}}{\partial s_m^i} * \frac{\partial s_m^i}{\partial o_m^i} + \sum_{j!=i}^{C} \frac{\mathcal{L}_{KL}}{\partial s_m^j} * \frac{\partial s_m^j}{\partial o_m^i}$$

$$= (\log s_m^i - \log s_n^i + 1) * s_m^i * \sum_{j!=i}^{C} s_m^j + \sum_{j!=i}^{C} (\log s_m^j - \log s_n^j + 1) * (s_m^j * s_m^i)$$

$$= \sum_{i!=j}^{C} ((\log s_m^i - \log s_m^j) - (\log s_n^i - \log s_n^j)) * (s_m^j * s_m^i)$$

$$= \sum_{i!=j}^{C} ((o_m^i - o_m^j) - (o_n^i - o_n^j)) * (s_m^j * s_m^i)$$

$$= \sum_{i!=j}^{C} (\Delta m_{i,j} - \Delta n_{i,j}) * w_m^{i,j}$$

$$= \sum_{j}^{C} (\Delta m_{i,j} - \Delta n_{i,j}) * w_m^{i,j} \tag{12}$$

For DKL los, we have the following derivatives according to the chain rule:

$$\frac{\partial \mathcal{L}_{DKL}}{\partial o_n^i} = \beta * s_m^i * (s_n^i - 1) + s_n^i * (1 - s_m^i)$$

$$\frac{\partial \mathcal{L}_{DKL}}{\partial o_m^i} = \frac{\alpha}{4} * 2 * (\sum_{j}^{C} (\Delta m_{j,i} - \Delta n_{j,i}) * (-w_m^{j,i}) + \sum_{k}^{C} (\Delta m_{i,k} - \Delta n_{i,k}) * w_m^{i,k})$$

$$= \alpha * \sum_{j}^{C} (\Delta m_{i,j} - \Delta n_{i,j}) * w_m^{i,j} \tag{13}$$

Combining with Appendix A.1, we claim that DKL loss and KL loss enjoy the same derivatives give the same inputs. Thus, DKL loss is equivalent to KL loss in training optimization.

Table 6: **Top-1 accuracy (%) on the CIFAR-100 validation.** Teachers and students are in **different** architectures. And $\Delta$ represents the performance improvement over the classical KD. All results are the average over 3 trials.

| Distillation Manner | Teacher Student | ResNet32×4 79.42 ShuffleNet-V1 70.50 | WRN-40-2 75.61 ShuffleNet-V1 70.50 | VGG13 74.64 MobileNet-V2 64.60 | ResNet50 79.34 MobileNet-V2 64.60 | ResNet32×4 79.42 ShuffleNet-V2 71.82 |
|---|---|---|---|---|---|---|
| Features | FitNet | 73.59 | 73.73 | 64.14 | 63.16 | 73.54 |
| | RKD | 72.28 | 72.21 | 64.52 | 64.43 | 73.21 |
| | CRD | 75.11 | 76.05 | 69.73 | 69.11 | 75.65 |
| | OFD | 75.98 | 75.85 | 69.48 | 69.04 | 76.82 |
| | ReviewKD | **77.45** | 77.14 | 70.37 | 69.89 | **77.78** |
| Logits | DKD | 76.45 | 76.70 | 69.71 | 70.35 | 77.07 |
| | KD | 74.07 | 74.83 | 67.37 | 67.35 | 74.45 |
| | **IKL-KD** | 76.64 ± 0.02 | **77.19** ± 0.01 | **70.40** ± 0.03 | **70.62** ± 0.08 | 77.16 ± 0.04 |

Table 7: **New state-of-the-art on public leaderboard** *R*obustBench Croce & Hein (2020).

| Experimental Settings | augmentation strategy | Clean | AA | Computation saving |
|---|---|---|---|---|
| w/o Generated Data (Previous best results) | Basic | 62.99 | 31.20 | |
| w/o Generated Data (Ours) | Basic | **64.08** | **31.65** | **33.3%** |
| w/o Generated Data (Previous best results) | Autoaug | **68.75** | 31.85 | |
| w/o Generated Data (Ours) | Autoaug | 64.63 | **32.52** | **33.3%** |
| w/ Generated Data (Previous best results) | Genreated data | 72.58 | 38.83 | |
| w/ Generated Data (Ours) | Generated data | **73.85** | **39.18** | 0% |

Table 8: **Comparisons with strong training settings on ImageNet for knowledge distillation.**

| Method | Top-1 |
|---|---|
| KD | 80.89 |
| DKD | 80.77 |
| DIST | 80.70 |
| IKL-KD | 80.98 |

## A.2  NEW STATE-OF-THE-ART ROBUSTNESS ON CIFAR-10/CIFAR-100

Robustbench is the most popular benchmark for adversarial robust models in the community. It evaluates the performance of models by the auto-attack. Auto-attack Croce & Hein (2020) is an ensemble of different kinds of attack methods and is considered the most effective method to test the robustness of models.

We achieve new state-of-the-art robustness on CIFAR-10 and CIFAR-100 under both settings w/ and w/o generated data. As shown in Table 7, on CIFAR-100 without extra generated data, we achieve 32.52% robustness, outperforming the previous best result by **0.67%** while saving **33.3%** computational cost. With generated data, our model boosts performance to 73.85% natural accuracy, surpassing the previous best result by **1.27%** while maintaining the **strongest robustness**. More detailed comparisons can be accessed on the public leaderboard https://robustbench.github.io/.

## A.3  MORE COMPARISONS ON CIFAR-100 FOR KNOWLEDGE DISTILLATION

We experiment on CIFAR-100 with the case that the teacher and student models have different unit network architectures. The results are listed in Table 6.

We follow the concurrent work Hao et al. (2023) and conduct experiments with BEiT-Large as the teacher and ResNet-50 as the student under a strong training scheme, the experimental results are summarized in Table 8. The model trained by IKL-KD shows slightly better results.

Table 9: **Ablation study of hyper-parameters $\alpha$ and $\beta$ in IKL.**

| $\alpha$ | Clean | AA | APGD-CE | APGD-T | | $\beta$ | Clean | AA | APGD-CE | APGD-T |
|---|---|---|---|---|---|---|---|---|---|---|
| $\alpha = 12$ | 67.24 | 30.64 | 34.46 | 30.64 | | $\beta = 1$ | 66.68 | 30.69 | 34.22 | 30.66 |
| $\alpha = 16$ | 66.60 | 30.72 | 34.43 | 30.72 | | $\beta = 2$ | 66.56 | 30.80 | 34.70 | 30.80 |
| $\alpha = 20$ | 66.51 | 31.45 | 35.46 | 31.45 | | $\beta = 3$ | 66.51 | 31.45 | 35.46 | 31.45 |
| $\alpha = 24$ | 63.59 | 31.44 | 35.65 | 31.45 | | $\beta = 4$ | 65.45 | 31.08 | 35.44 | 31.08 |

(a) Effects of $\alpha$ on adversarial robustness.      (b) Effects of $\beta$ on adversarial robustness.

Table 10: **Ablation study of temperature $\tau = 4$ for global information.**

| Hyper-parameter $\tau = 4$ with $\alpha$ | Clean | AA | | Hyper-parameter $\tau = 4$ with $\beta$ | Clean | AA |
|---|---|---|---|---|---|---|
| $\alpha = 12$ | 67.24 | 30.64 | | $\beta = 1$ | 66.68 | 30.69 |
| $\alpha = 16$ | 66.60 | 30.72 | | $\beta = 2$ | 66.56 | 30.80 |
| $\alpha = 20$ | 66.51 | **31.45** | | $\beta = 3$ | 66.51 | **31.45** |
| $\alpha = 24$ | 63.59 | 31.44 | | $\beta = 4$ | 65.45 | 31.08 |

(a) Effects of $\alpha$ with $\beta = 3$.      (b) Effects of $\beta$ with $\alpha = 20$.

Table 11: **Ablation study of temperature $\tau = 2$ for global information.**

| Hyper-parameter $\tau = 2$ with $\alpha$ | Clean | AA | | Hyper-parameter $\tau = 2$ with $\beta$ | Clean | AA |
|---|---|---|---|---|---|---|
| $\alpha = 15$ | 65.12 | 31.17 | | $\beta = 2$ | 64.30 | 31.46 |
| $\alpha = 18$ | 64.63 | 31.34 | | $\beta = 3$ | 64.31 | 31.59 |
| $\alpha = 20$ | 64.31 | **31.59** | | $\beta = 4$ | 64.08 | **31.67** |
| $\alpha = 24$ | 63.59 | 31.44 | | $\beta = 5$ | 63.58 | 31.62 |

(a) Effects of $\alpha$ with $\beta = 3$.      (b) Effects of $\beta$ with $\alpha = 20$.

## A.4 ABLATIONS FOR ADVERSARIAL ROBUSTNESS

**Hyper-parameters of $\alpha$ and $\beta$.** With IKL, the two components can be manipulated independently. We empirically study the effects of hyper-parameters of $\alpha$ and $\beta$ on CIFAR-100 for adversarial robustness. Robustness under APGD-CE, APGD-T, and AA Croce & Hein (2020) are reported in Table 11. Especially, only samples that can not be attacked by APGD-CE will be tested under APGD-T attack. Reasonable $\alpha$ and $\beta$ should be chosen for the best trade-off between natural accuracy and adversarial robustness.

**Ablation study of temperature for global information.** As described in Section 3.2, corporating global information, the class-wise weights is proposed to promote intra-class consistency and mitigate the biases from sample noise,

$$\bar{w}_y^{j,k} = \bar{s}_y^j * \bar{s}_y^k, \tag{14}$$

where $y$ is ground-truth label of $x_m$, $\bar{s}_y = \frac{1}{|\mathcal{X}_y|} \sum_{x_i \in \mathcal{X}_y} s_i$.

We further examine the effect of temperature $\tau$ and extend the class-wise weights as,

$$\bar{w}_y^{j,k} = \bar{s}_y^j * \bar{s}_y^k, \tag{15}$$

where $y$ is ground-truth label of $x_m$, $\bar{s}_y = \frac{1}{|\mathcal{X}_y|} \sum_{x_i \in \mathcal{X}_y} s_i$, and $s_i = Softmax(o_i/\tau)$.

**Ablation Study of Robustness under Different Perturbation Size** Auto-attack is an ensemble of different attack methods, including APGD-CE, APGD-DLR, FAB, and Square Attack. It is the most popular benchmark for evaluating the adversarial robustness of models (`https://robustbench.github.io/`).

We train models with IKL-AT and Improved Trades on CIFAR-100. The same experimental settings are adopted. we train the models 200 epochs and use the perturbation size of 8/255 for generating the adversarial examples during training. The evaluation under different perturbation sizes is listed in Table 12. Our model trained by IKL-AT consistently outperforms the baselines.

Table 12: **Ablation study of robustness under different perturbation sizes.**

| Method | Epsilon | AA |
|---|---|---|
| Improved Trades | 2/255 | 53.88 |
| IKL-AT | 2/255 | 55.31 |
| Improved Trades | 4/255 | 45.31 |
| IKL-AT | 4/255 | 46.76 |
| Improved Trades | 6/255 | 37.28 |
| IKL-AT | 6/255 | 38.98 |
| Improved Trades | 8/255 | 30.29 |
| IKL-AT | 8/255 | 31.67 |
| Improved Trades | 10/255 | 24.28 |
| IKL-AT | 10/255 | 25.33 |
| Improved Trades | 12/255 | 19.17 |
| IKL-AT | 12/255 | 19.98 |

## A.5 CODE AND PRE-TRAINED MODELS

On adversarial training with CIFAR-100 and CIFAR-10, we achieve the new state-of-the-art in both settings with/without data augmentations. Our pre-trained models are available to be evaluated.

- CIFAR-100 (clean 66.51 AA 31.45): https://drive.google.com/file/d/1GzRey51JGmYNZTV79M_qHCL03tIf6X1P/view?usp=sharing

- CIFAR-100 (clean 63.40 AA 31.92): https://drive.google.com/file/d/1iB31b5bGyLbotQMrwd7A2nlrjKH9uO9l/view?usp=drive_link

- CIFAR-100 (clean 73.85 AA 39.18): https://drive.google.com/file/d/1Leec2X9kGBnBSuTiYytdb4_wR50ibTE8/view?usp=sharing

- CIFAR-10 (clean 85.31 AA 57.13): https://drive.google.com/file/d/1SFdNdKE6ezI6OsINWX-h74dGo2-9u3Ac/view?usp=sharing

- CIFAR-10 (clean 92.16 AA 67.75): https://drive.google.com/file/d/1gEodZ4ushbRPaaVfS_vjJyldH3wJg4zV/view?usp=sharing

- Evaluation code and logs with auto-attack: https://drive.google.com/file/d/1W96kAkGIiY4aCD9YKxPQogI3K2FEzHiH/view?usp=sharing