# OpenReview forum: "Decoupled Kullback-Leibler Divergence Loss"
_ICLR.cc/2024/Conference — ICLR 2024 Conference Withdrawn Submission_

### Official Review · Reviewer_2ej3 · 2023-11-01

**Soundness:** 3 good
**Presentation:** 2 fair
**Contribution:** 3 good
**Rating:** 5
**Confidence:** 4

**Summary:**

The paper aims to improve the KL divergence loss in the context of Adversarial Training and Knowledge Distillation. Based on gradient perspective, the authors prove KL divergence can be decomposed into weighted Mean Square Error (wMSE) and Cross Entropy loss with soft labels. Then, the authors propose two improvements by i) breaking the asymmetric property by allowing the gradient back-propagation through student in wMSE loss and ii) introducing the global information by substituting the individual-wise weights samples by class-wise weights in wMSE to mitigate the bias of per-sample. Experiments with CIFAR-10, CIFAR-100 and ImageNet datasets on adversarial robustness and knowledge distillation to support the claims.

**Strengths:**

Overall, the decomposition of KL in gradient perspective and the proposed improvements seem novel to me. The results are encouraging compared to few baseline methods.

**Weaknesses:**

However, there is some weakness in writing the Experimental section which makes it hard to understand a few figures. The results seem incremental between DKD and IKL and it would be good to have more ablation studies to understand the impacts of individual components.

**Questions:**

Is there a typo in Eq. 10? Do the authors need to keep the gradient stopping on the teacher’s outputs in wMSE like in Eq. 7?

I am curious regarding the nature of asymmetries of KL divergence. Instead of improving by breaking this why don’t just use Jensen Shannon divergence instead?

Also, it would be beneficial to provide experiments from where to motivate the authors to figure out the weakness of the symmetric breaking?

Table 1 does not show GI plays an important role as it is only shown when combined with BA. What happens if GI is enabled alone? It would be beneficial to study this option as well to support the claim. This study is on AT, it would be beneficial to investigate for KD as well.

The experiments are not well-written when it does not provide sufficient information for understanding the experiments from the beginning.  In the first analysis of GI, what is IKL, IKL-AT? Why IKL-based methods and TRADES used for the experiments. They should be discussed in a bit more detail before the analysis. I just realized IKL-AT and IKL-KD are the same method applied for two different problems only when reading to the end. It would be beneficial that authors provide more details on comprehending the Fig. 2a? What TRADES play a role here? Would be good to refer the names of baseline methods such as LBGAT, AWP, etc into original papers to follow more easily.

What is the difference between KD and DKD in Table 5, why do the results are different as in theory they are equivalent?

---

### Official Review · Reviewer_6Mi2 · 2023-11-02

**Soundness:** 3 good
**Presentation:** 2 fair
**Contribution:** 2 fair
**Rating:** 5
**Confidence:** 2

**Summary:**

The paper delved into KL divergence loss and mathematically show that it is equivalent to DKL loss. The work then derive IKL loss and evaluate its effectiveness on varous datasets, focusing on knowledge  distillation and adversarial training.

**Strengths:**

- The paper studies one of the most important and interesting objective function in deep learning.
- The paper sections are generally well written.

**Weaknesses:**

- Notation-wise, I would suggest the authors to polish Section 3 to improve readability. The current paragraphs are a bit vague and unclear. For example, some of the notations are not properly introduced.

**Questions:**

- To what extent does the authors believe the proposed IKL loss can improve other domains in deep learning?
- From Table.1, it seems that GI loss is the most effective component in the new IKL objective, I wonder if this phenonemon is observed across all experiment settings?
- As I am not from either KD or Adversarial training community, although the theoritical derivation looks interesting, the empirical result does not look as exciting as the theoritical insights. I hope the authors can provide more insight on this.

---

### Official Review · Reviewer_MJfm · 2023-11-04

**Soundness:** 2 fair
**Presentation:** 2 fair
**Contribution:** 2 fair
**Rating:** 3
**Confidence:** 4

**Summary:**

This paper delves deeper into the Kullback-Leibler (KL) Divergence loss and mathematically proves that it is equivalent to the Doupled Kullback-Leibler (DKL) Divergence loss that consists of 1) a weighted mean square error (wMSE) loss and 2) a cross-entropy loss incorporating soft labels. From the analysis of the DKL loss, two areas for improvement are identified. First, the limitation of DKL in scenarios like knowledge distillation by breaking its asymmetry property in training optimization is addressed. Second, global information is introduced into DKL to mitigate bias from individual samples. With these two enhancements, the improved Kullback-Leibler is presented. Experiments on CIFAR-10/100 and ImageNet datasets, focusing on adversarial training and knowledge distillation tasks. The proposed approach achieves new state-of-the-art adversarial robustness on the public leaderboard.

**Strengths:**

- The motivation of this paper is overall clear.
- The organization of this paper is good.

**Weaknesses:**

- The technical contribution is limited for top-tier conferences.
- Empirical evaluations are not sufficient and convincing. More experiments should be supplemented.
- Writing needs to be polished further to make the content easy to understand.

More details about weaknesses are provided below.

**Questions:**

- The paper claims that although many of these studies incorporate KL Divergence loss as part of their algorithms, they may not thoroughly investigate the underlying mechanisms of the loss function. I am confused about this. As the KL Divergence loss is not new, many topics use this loss. Why there is no study for this loss?
- Could the paper detail the claim that its gradient optimization is asymmetric?
- Figure 1 is not very informative. From this figure, the issue of asymmetric gradient optimization is not reflected.
- This paper mainly studies the property of KL loss. Adv. robustness can be seen as an application. I am confused about why this paper reviews lots of works on adv. robustness. It seems that these works are not very related to the core content of this work.
- Could the paper provide a formal definition of stop gradients operation? It will help readers understand the theoretical analysis more clearly. Also, could the paper provide some intuitions on how to analyze the loss form but with gradient information?
- Potentially, could the modified robust loss function break the previous nice character of KL loss?
- The difference between Figure 2(b) and Figure 2(c) is not very clear. Also, the improvement brought by the proposed method is marginal in lots of cases. Besides, the use of the proposed method may bring side effects on final performance, which is worrying.
- Could the proposed method of this paper be applied to the SOTA KD methods for better performance? The experiments about current baselines are not very convincing.

---

### Official Review · Reviewer_JUiH · 2023-11-08

**Soundness:** 3 good
**Presentation:** 2 fair
**Contribution:** 2 fair
**Rating:** 3
**Confidence:** 4

**Summary:**

The paper examines the Kullback–Leibler (KL) Divergence loss, a measure of dissimilarity between probability distributions, and demonstrates its mathematical equivalence to the Decoupled Kullback-Leibler (DKL) Divergence loss. The DKL combines a weighted Mean Square Error (wMSE) with a Cross-Entropy loss that uses soft labels. Two enhancements to the DKL are proposed: (1) breaking its asymmetry to improve its efficiency during training, and (2) adding global information to reduce sample bias. These improvements lead to the creation of the Improved Kullback–Leibler (IKL) Divergence loss. The effectiveness of the IKL loss is validated through adversarial training and knowledge distillation tasks on CIFAR-10/100 and ImageNet datasets.

**Strengths:**

1. The paper is easy to understand and follow.
2. In addition to empirical results, the authors also provide mathematical and theoretical guarantees.
3. The results on adversarial robustness and knowledge distillation somehow show the effectiveness of the proposed method.

**Weaknesses:**

1. The paper lacks a clear explanation for the selection of the two tasks in their experiments. Since KL loss is widely used in many problems in the field of ML and beyond, it is unclear whether these tasks were randomly chosen or if the proposed method is particularly suited for these specific tasks. Additionally, the effectiveness of the proposed method on image classification does not indicate it is also effective when applying to more challenging KD problems, such as object detection. It would be better to clarify and evidence the capability of the proposed method.

2. The paper contains certain terms that are not clearly defined, for instance, "hard examples." Could you please clarify what constitutes a "hard example"? Are there any sources you could cite to support this definition? Does this term refer solely to samples that the model frequently misclassifies, or does it also include instances that the model predicts correctly but with low confidence?

3. The paper contains too many typos, inconsistencies, and format issues, including but not limited to:

(1) In the abstract: "Doupled" --> "Decoupled". (This is significant since this is the first time this key term apprears in the paper.)

(2) Page 6, last paragraph: "Features" --> "features"

(3) The caption of Table 5: "And Δ represents the performance improvement over the classical KD." Where is Δ in the table?

(4) Below Table 5: "We" --> "we"

(5) Last sentence in "Conclusion": "we" --> "We"

(6) The author should use "\citep" for including references to make the format as (Author et al., 2015; Author et al., 2022; Author
et al., 2019), rather than Author et al. (2015); Author et al. (2022); Author et al. (2019).

**Questions:**

1. For breaking the asymetry property, why do not choose to use JS divergence? What is the relation and difference between the proposed modification compared to JS divergence, both mathematically and empirically?
2. For Table 4, if consideirng the performance both on Clean and AA, it is not clear to see the benefit of IKL-AT. For instance, comparing ACAT and IKL-AT in the middle block, it seems ACAT performs better. What is the tradeoff here?